# Sex Differences in Expression of Pro-Inflammatory Markers and miRNAs in a Mouse Model of CVB3 Myocarditis

**DOI:** 10.3390/ijms25179666

**Published:** 2024-09-06

**Authors:** Misael Estepa, Maximilian H. Niehues, Olesya Vakhrusheva, Natalie Haritonow, Yury Ladilov, Maria Luisa Barcena, Vera Regitz-Zagrosek

**Affiliations:** 1Department of Internal Medicine and Cardiology, Deutsches Herzzentrum der Charité, 13353 Berlin, Germany; 2Institute for Gender in Medicine, Center for Cardiovascular Research, Charité University Hospital, 10115 Berlin, Germany; maxniehues@outlook.de (M.H.N.); vera.regitz-zagrosek@charite.de (V.R.-Z.); 3Department of Urology, University Hospital Tübingen, 72076 Tübingen, Germany; 4Department of Geriatrics and Medical Gerontology, Charité—Universitätsmedizin Berlin, Corporate Member of Freie Universität Berlin and Humboldt–Universität zu Berlin, 12203 Berlin, Germany; 5Heart Center Brandenburg, Department of Cardiovascular Surgery, Brandenburg Medical School, 16321 Bernau bei Berlin, Germany; yury.ladilov@rub.de; 6Department of Cardiology, University Hospital Zürich, University of Zürich, 8091 Zürich, Switzerland

**Keywords:** sex differences, estrogen, cardiac inflammation, microRNAs, CVB3-induced chronic myocarditis

## Abstract

Myocarditis is an inflammatory disease that may lead to dilated cardiomyopathy. Viral infection of the myocardium triggers immune responses, which involve, among others, macrophage infiltration, oxidative stress, expression of pro-inflammatory cytokines, and microRNAs (miRNAs). The cardioprotective role of estrogen in myocarditis is well documented; however, sex differences in the miRNA expression in chronic myocarditis are still poorly understood, and studying them further was the aim of the present study. Male and female ABY/SnJ mice were infected with CVB3. Twenty-eight days later, cardiac tissue from both infected and control mice was used for real-time PCR and Western blot analysis. NFκB, IL-6, iNOS, TNF-α, IL-1β, MCP-1, c-fos, and osteopontin (OPN) were used to examine the inflammatory state in the heart. Furthermore, the expression of several inflammation- and remodeling-related miRNAs was analyzed. NFκB, IL-6, TNF-α, IL-1β, iNOS, and MCP-1 were significantly upregulated in male mice with CVB3-induced chronic myocarditis, whereas OPN mRNA expression was increased only in females. Further analysis revealed downregulation of some anti-inflammatory miRNA in male hearts (let7a), with upregulation in female hearts (let7b). In addition, dysregulation of remodeling-related miRNAs (miR27b and mir199a) in a sex-dependent manner was observed. Taken together, the results of the present study suggest a sex-specific expression of pro-inflammatory markers as well as inflammation- and remodeling-related miRNAs, with a higher pro-inflammatory response in male CVB3 myocarditis mice.

## 1. Introduction

Viral myocarditis represents an inflammatory state of the myocardium triggered, i. a. by viral infection [1]. Acute viral myocarditis may progress to chronic ventricular dysfunction, e.g., dilated cardiomyopathy [2]. Among the identified viruses leading to viral myocarditis, enteroviruses, particularly coxsackie B viruses, account for 25% of the cases [2]. Several mouse models have been used to study the development and progress of acute and chronic viral myocarditis. While C57BL/6 mice do not develop inflammatory dilated cardiomyopathy (DCMI) or DCM, mice with BALB/c, A/J, and ABY/SnJ genetic backgrounds develop chronic myocarditis 28 days after infection, leading to myocardial fibrosis and dilated cardiomyopathy [3,4,5,6].

The immune response triggered by viral infection plays a key role in myocardial damage induced by CVB3 [7]. Among others, natural killer cells, macrophages, and T lymphocytes, as well as pro-inflammatory cytokines such as interleukins (e.g., IL-6 or IL-1β), TNF-α, and IFN-α, play a fundamental role in the activation of antiviral immune responses [7]. 

Sex differences in myocarditis have been shown, with a higher prevalence in men as well as worse clinical presentations, stronger inflammatory response, and worse outcomes [8]. The cardioprotective effects observed in young women are attributed to estrogen and estrogen receptors [9]. Similarly to patients, male mice show a greater inflammatory response than females, which correlates with the disease severity [10,11]. Particularly, male mice with CVB3-induced myocarditis display a higher Th1 response and release of pro-inflammatory cytokines such as IL-1β, IL-18, and IFN-γ, whereas female mice develop predominantly an anti-inflammatory Th2 response in the context of CVB3-induced myocarditis [12]. These sex differences may be due to the estradiol-dependent modulation of immune cells, via estrogen receptors, which have been shown to regulate the gene expression of key mediators related to differentiation of immune cell precursors and activation of pro- and anti-inflammatory pathways in mature cells, such as Type I Interferon, TNFα, and NF-κB [13]. In addition, estradiol (E2) hinders the expression of pro-inflammatory cytokines and adhesion molecules and boosts phagocytosis in these cells in females [14].

Emerging evidence suggests the role of microRNAs (miRNAs) in acute myocarditis [15]. miRNAs are endogenous, small, conserved, non-coding RNAs that regulate gene expression post-transcriptionally, and thus are essential epigenetic regulators of cardiovascular physiology and pathophysiology [16]. Several miRNAs, e.g., miR21, miR1, miR20b, miR126, miR155, and miR-125b, are involved in the inflammatory reaction after acute viral myocardial infection [7,15]. However, the role of miRNAs in chronic myocarditis is still obscure. Importantly, dysregulation of miRNA expression observed during acute viral myocarditis also has been demonstrated in chronic viral myocarditis, suggesting their contribution to remodeling and fibrosis [17,18]. A dysregulated miRNA expression might result in myocarditis aggravation or amelioration [15]. It has also been shown that silencing some of these miRNAs induces anti-inflammatory M2 macrophage polarization, leading to the attenuation of cardiac injury during CVB3-induced myocarditis [19,20]. Furthermore, miRNAs play a role in cardiac remodeling, particularly via the regulation of myocardial fibrosis, as they have been shown to interact with key fibrosis-related mediators such as TGF-β1 and TGF-βRII [15]. For example, miR21 may promote cardiac fibrosis via the upregulation of TGF-β1 in a mouse model of chronic viral myocarditis [21]. 

In this study, we aimed to examine the sex-specific differences in the inflammatory response and regulation of the inflammation- and remodeling-related miRNAs in a mouse model of chronic viral myocarditis. 

## 2. Results

### 2.1. CVB3-Infected Mice Show Cardiac Hypertrophy, Increased Immune Infiltrates and Fibrosis in Myocardial Tissue

Twenty-eight days after CVB3 myocardial infection, significant cardiac hypertrophy, as defined by the increased heart to body weight ratio (Figure 1A), was observed in male and female mice. Male mice, however, showed a significant reduction in body (Figure 1B) and spleen (Appendix A) weight compared to female mice (Figure 1B and Appendix A). The relative spleen weight to body weight ratio did not vary in the chronic myocarditis mice (*p* > 0.05) (Appendix A). 

Hematoxylin/eosin staining revealed an increased number of cell infiltrates in the myocardial tissue of male and female CVB3-infected mice (*p* = 0.0105 and 0.0221, respectively) (Figure 1D,E). In accordance with the increased number of infiltrates, male and female mice with CVB3-induced chronic myocarditis showed higher amounts of pathological fibrosis in the myocardium in comparison to uninfected mice. However, only male mice showed a significant accumulation of fibrotic tissue (*p* = 0.0070) (Figure 1F,G). Col1A1 and Col3A1 expression was not affected by CVB3 infection (Figure 1H,I). Moreover, the OPN mRNA expression was significantly higher in CVB3-myocarditis females (*p* = 0.0127) (Figure 1J).

To further characterize the cell infiltrates observed in CVB3-activated chronic myocarditis in mice, staining with CD68, CD11b, and CD8 antibodies was performed. CD68 and CD11b, valuable markers of monocytes/macrophages in inflamed tissue, were significantly elevated in heart tissue of CVB3-treated male mice compared to untreated control (*p* = 0.002 and 0.0018, respectively) (Figure 2A–C). In female mice, increased CD68 and CD11b expression was observed but did not reach statistical significance (Figure 2A–C). In contrast, no differences in the expression of CD8, a marker for cytotoxic T cells, were determined in all groups investigated (Figure 2D).

### 2.2. Sex-Dependent Downregulation of ERK Activity in the Mouse Model of CVB3-Induced Chronic Myocarditis

It has been shown that ERK, p38, and AKT signaling pathways play a significant role in cardiac remodeling [22,23] and are affected by CVB3-induced myocarditis [24,25]. Furthermore, sex hormones can modulate p38 and ERK signaling and thus may contribute to sex differences in the inflammatory processes [26,27,28]. Therefore, we first analyzed the phosphorylation, and thus the activation, of these kinases in myocardial tissue from male and female mice infected with CVB3. ERK phosphorylation was significantly decreased in male but not in female CVB3 mice (*p* = 0.0101) (Figure 3A), while the p38 and AKT phosphorylation remained unaffected by CVB3 infection in both male and female mice (*p*> 0.05) (Figure 3B,C). 

The expressions of cardiac ERα and ERβ were similar in the CVB3-induced chronic myocarditis and control groups in both male and female mice (*p* > 0.05) (Appendix A).

### 2.3. Sex Differences in the Expression of Pro-Inflammatory Mediators in the Mouse Model of CVB3-Induced Chronic Myocarditis

We further examined whether the chronic myocarditis-related downregulation of ERK phosphorylation was accompanied by sex-dependent alterations in the inflammatory state. Of note, numerous pro-inflammatory markers were significantly elevated in male but not in female CVB3-infected mice, i.e., NF-kB and IL-6 proteins as well as IL-1β, iNOS, and MCP-1 RNAs (Figure 4A–E,I). In line with the reduced ERK phosphorylation in male CVB3-infected mice, the expression of the inflammation modulator c-fos was significantly reduced in male CVB3-infected mice (*p* = 0.0426) (Figure 4J). 

Similarly, the anti-inflammatory factor RELMα was decreased in CVB3-infected males in comparison to non-infected males, while female hearts expressed three times more RELMα (*p* = 0.0513 and 0.1206, respectively) (Figure 4K). No effects of chronic myocarditis on the expression of TLR4 or TGF-β have been found (*p* > 0.05) (Figure 4G–H,L). 

### 2.4. Sex Differences in the Inflammatory- and Remodeling-Related miRNA Expression in Mice with CVB3-Induced Chronic Myocarditis

miRNAs are involved in the regulation of cardiac function, the immune response/inflammation (e.g., let7a, let7b, let7d, and miR199a) [7,29,30,31,32,33], and cardiac remodeling (e.g., miR27b, miR199a) [34,35,36]. Moreover, the expression of some miRNAs, like miR27b, is affected by estradiol [37]. Therefore, in this study, we examined sex differences in the expression of some miRNAs that are known to be involved in cardiac function, inflammation, fibrosis, and remodeling. In the CVB3-induced myocarditis group, the expression of the anti-inflammatory miRNA let7a was reduced compared to the control group in a sex-independent manner (*p* = 0.0016 and 0.0549) (Figure 5A). let7b, let7d, and miR27b were not affected by chronic myocarditis in male mice (*p* > 0.05), while the anti-inflammatory miRNA let7b and the remodeling-related miRNA miR27b were upregulated in females infected with CVB3 (*p* = 0.0278 and *p* = 0.0295, respectively) (Figure 5B,D). It is important to note that sex differences in the expression of let7a, let7b, and let7d were observed in the control group, with significantly higher expression in male mice (*p* = 0.0051, *p* = 0.0549 and *p* = 0.0133, respectively) (Figure 5A–C). The miR199a expression was significantly reduced only in male CVB3-infected mice (*p* = 0.0186) (Figure 5E).

## 3. Discussion

The present study aimed to investigate sex-dependent differences in the expression of inflammatory mediators and relevant miRNAs involved in inflammatory processes and cardiac remodeling in cardiac tissue from male and female ABY/SnJ mice with and without CVB3-induced chronic myocarditis. The main findings in this study are as follows: (1) CVB3-induced chronic myocarditis was accompanied by myocardial immune cell infiltrates and pathological fibrosis formation as well as a downregulation of ERK/c-fos signaling and up-regulation of pro-inflammatory markers (e.g., NFκB, IL-6, IL-1β, iNOS) in a sex-dependent manner; (2) CVB3-induced chronic myocarditis led to dysregulation of the expression of several inflammation- and remodeling-related miRNAs; some of them were specifically upregulated in female (let7b and miR27b) mice or downregulated in male (miR199a) mice.

Sex-dependent differences in the incidence, progression, and outcomes of myocarditis are well-documented, as it is known that myocarditis affects males more frequently than females and is characterized by increased inflammation [8,10], which may be related to the cardioprotective effects of estrogen [9,38]. The underlying cellular mechanisms of the sexual dimorphism, however, are still poorly understood. In the current study, we used ABY/SnJ mice as a model of chronic myocarditis, which developed 28 days after CVB3 infection. The model has been well-characterized in previous studies and is accompanied by myocardial inflammation and fibrosis, leading to dilated cardiomyopathy [3,4,5,6]. Accordingly, our study demonstrates an increased number of immune cell infiltrates in the myocardium of male and female CVB3-induced chronic myocarditis mice. Likewise with this finding, in EAM rats, myocardial immune infiltrates were increased in males and females [39], suggesting that both sexes developed an immune response due to CVB3 infection.

It is well-established that MAP kinases play a role in CVB3-induced myocarditis [24,25], and CVB3 myocardial infection may affect ERK signaling [40], because activation of ERK is relevant for the replication of CVB3 [41,42]. In the present study, we found a decline in ERK phosphorylation in male CVB3-induced myocarditis mice, whereas the phosphorylation of other investigated kinases was not affected. It has been reported that sex hormones, like estrogen and testosterone, stimulate MAP kinase signaling via membrane-associated receptors [43,44], and differences in the phosphorylation of MAPK between females and males in skeletal muscle have been shown [45]. It is particularly interesting to note that several studies assume testosterone metabolites act in a neuroprotective manner due to the inhibition of ERK phosphorylation in neurons and neuroblastoma cells during neurotoxic processes [46,47], suggesting that the decreased phosphorylation rate, and therefore the ERK activity in males during viral-induced chronic myocardial inflammation, may be testosterone-dependent and may have protective functions. The pro-inflammatory actions of c-fos are associated with myocardial damage and cell infiltration in the CVB3 myocarditis mouse model [48]; however, in our study, c-fos expression was downregulated in the hearts of male mice with chronic myocarditis, suggesting an anti-inflammatory effect in chronic myocarditis, since c-fos also promotes anti-inflammatory actions [49]. In accordance, the anti-inflammatory factor RELMα was also significantly downregulated in male mice with chronic myocarditis. 

Key pro-inflammatory mediators like NFκB, TNF-α, and IL-1β are upregulated in mouse models of viral myocarditis and autoimmune myocarditis and play a crucial role in myocarditis progression [39,50,51,52]. In line with other studies [50,53], we found significantly increased pathological fibrotic tissue in the hearts of male mice with chronic myocarditis. The increased immune cell infiltrates and pathological fibrosis formation in our study seem to be linked to the upregulation of NFκB, IL-6, TNF-α, IL-1β, and iNOS specifically in the male chronic myocarditis group, which are detrimental when released in an excessive quantity [52,54]. Similarly, MCP-1, which is important for macrophage function and inflammation [55] and is overexpressed during myocarditis [56], was also increased specifically in male mice with chronic myocarditis. Similarly with other studies [39,57,58,59] that found a more severe inflammatory response in males in acute or chronic myocarditis, our study revealed a lack of pro-inflammatory response in female mice with chronic myocarditis. The pro-inflammatory and pro-fibrotic [60] cytokine OPN is released mainly by macrophages, fibroblasts, and T cells under pathological conditions such as myocarditis, coronary heart disease, myocardial infarction, and hypertrophy [61]. In addition, it has been suggested that its high concentrations may correlate with the induction of myocardial fibrosis [61,62]. In agreement with these studies, Szalay et al. demonstrated an increased OPN expression in CVB3-mediated myocarditis mice, which correlated with a susceptibility to the development of chronic myocarditis [61]; however, the sex difference was not investigated. In our study, OPN was only upregulated in female mice after chronic CVB3 infection. Thus, we suggest that the sex-dependent pattern of pro-inflammatory markers’ expression may be, in part, responsible for the previously shown exacerbated inflammatory response leading to worse outcomes in males [39]. 

Emerging evidence suggests that miRNA is a key player in inflammatory processes and particularly in viral myocarditis [7]. miRNAs are small non-coding RNAs that regulate gene expression post-transcriptionally, playing an important role in cellular function and disease [16]. The let7 family belongs to the miRNAs related to cardiovascular physiology and disease [29]. Let7a is known to regulate inflammation and apoptosis [30]. Particularly, let7a seems to have anti-inflammatory effects, since its overexpression in macrophages leads to the downregulation of pro-inflammatory cytokines, including TNF-α and IL-6, and upregulation of the anti-inflammatory cytokines IL-10 and IL-4 [30]. In the current study, we observed a significant decrease in let7a in male myocarditis mice, which may at least partly explain the upregulation of several pro-inflammatory mediators specifically in males. In addition to let7a, let7b and let7d may also have anti-inflammatory effects [31,32,63]. Similarly to let7a, these miRNAs show a tendency to downregulate in male mice, whereas let7b increased significantly in female mice. Interestingly, we found sex differences in the expression of let7a, let7b, and let7d at baseline levels, as they were shown to be more expressed in males. Thus, sex difference in the dysregulation of anti-inflammatory miRNAs may be responsible for the upregulation of pro-inflammatory mediators specifically in male mice with chronic myocarditis.

The increased OPN expression in female CVB3-induced chronic myocarditis mice correlates with the elevated expression of the remodeling-related miRNA, miR27b, in the hearts of female mice with chronic myocarditis, suggesting that females are more susceptible to remodeling after chronic CVB3 infection.

A variety of factors may contribute to the sex difference in the pathogenesis of CVB3-induced myocarditis (for review see [64]). Gonadal hormones seem to play a primary role here. In particular, gonadectomy in mice suppressed male, but increased female, susceptibility to CVB3 [65]. The protective effects of the female hormone estradiol and estrogen receptors due to the affected pro- and anti-inflammatory pathways in immune cells have been previously shown [13,14]. In contrast to estrogen, the male sex hormone testosterone may contribute to the worse myocarditis outcome in males [8]. Finally, sex chromosomes may also affect the CVB3 pathogenesis [65,66]. Although the role of sex was poorly understood, the involvement of *Midline 1*, *TLR7* and *TLR8* genes localized on the X chromosome in immune response has been suggested [67,68,69].

### Limitations of the Study

In this study, a small animal cohort was investigated in order to contribute to the 3R- rules, i.e., replacement, reduction, and refinement, aimed at reducing the number of animals used in experiments. 

In addition, due to the S2 housing conditions, we were not able to analyze the cardiac function of the animals to prove dilated cardiomyopathy in mice with chronic myocarditis. The lack of functional analysis in vivo did not allow us to make any conclusions regarding dilation or hypertrophy in our mouse model for chronic myocarditis. Also, with the frozen tissue sections, we were not able to prove the dilation of the heart. Although we were able to demonstrate cardiac hypertrophy based on the HW/BW ratio (Figure 1), these data should be taken with caution, as body weight was reduced in mice with chronic myocarditis.

In the present study, the immune cell infiltrates were not characterized, leaving unresolved the question of which immune cells were involved in the observed inflammatory response in our mouse model for chronic myocarditis. Moreover, some pro- and anti-inflammatory markers were investigated in the present study only at the mRNA level, which did not allow us to demonstrate their activity. 

## 4. Materials and Methods

### 4.1. Animals

Mice (5 to 6 weeks old), both male and female ABY/SnJ (n = 16), were infected intraperitoneally (i.p.) with 10^5^ plaque-forming units (PFU) of purified CVB3 (cardiotropic Nancy strain) [70,71]. Mice were obtained through in-house breeding from stock originally purchased from the Jackson Laboratory. The cycles of the female mice were synchronized 7 days prior to infection. Twenty-eight days after viral infection, the mice were euthanized by cervical dislocation without anesthesia (Appendix A). Hearts, spleens, livers, and both kidneys were excised and snap-frozen in liquid nitrogen and stored at −80 °C. Non-infected ABY/SnJ mice were used as a control (n = 16). Mice were housed in cages with controlled temperature and humidity on a 12 h light/12 h dark cycle, with free access to food and water in the Research Institutes for Experimental Medicine of the Charité–Universitätsmedizin Berlin. After infection, all mice were randomized.

All procedures were performed in accordance with the Guide for the Care and Use of Laboratory Animals published by the US National Institutes of Health and with the European Legislation for the Care and Use of Laboratory Animals (Directive 2010/63/EU), and were approved by local authorities (Landesamt für Gesundheit und Soziales, Berlin, approval number H0076-08).

### 4.2. Analysis of Heart Weight to Body Weight Ratio

Body weight (BW) was measured before euthanasia. After euthanasia, the hearts without atria were weighed, and the relative heart weight (HW) to body weight (BW) ratio (HW/BW) was calculated [39].

### 4.3. Analysis of Collagen Content in LV Cardiac Tissue

Picrosirius red staining was performed in paraffin-embedded mouse LV cardiac tissue sections (5 µm) for collagen content assessment [39]. Images were acquired with the Axiophot microscope (Zeiss, Jena, Germany). The overall fibrosis was determined via semiquantitative, visual evaluation and then averaged. All sections were blindly evaluated by three different investigators, as described in [24].

### 4.4. Analysis of Immune Cell Infiltrates in LV Cardiac Tissue 

Paraffin-embedded mouse LV cardiac tissue sections (5 µm) were stained with hematoxylin and eosin to obtain the number of immune cell infiltrates in the myocardium of CVB3-infected mice [39]. Images were acquired with the Axiophot microscope (Zeiss, Jena, Germany). The amount of immune cell infiltrates was quantified in 10 high power fields (hpf) (field of vision in x200 original magnification) and then averaged. All sections were blindly evaluated by three different investigators, as described in [24].

To characterize the infiltrated cells, CD68 (OriGene Technologies GmbH, Herford, Germany), CD11b (Novus Biologicals, Bio-Techne GmbH, Wiesbaden, Germany) and CD8 (Clone MA5, Invitrogen, Thermo Fischer Scientific, Darmstadt, Germany) antibodies were used. Briefly, after antigen retrieval, samples were incubated with 10% BSA/PBST to block the unspecific binding. Thereafter, samples were probed with a primary antibody (CD68 1:250, CD11b 1:250 and CD8 1:100 dilution) for 1 h at RT and then reacted with secondary antibodies (Cy2 AffiniPure Donkey anti-rabbit IgG or Alexa Flour 488-conjugated AffiniPure Goat anti-mouse IgG (both: Jackson ImmunoResearch Europe LTD, Biozol Diagnostica GmbH, Eching, Germany); both 1:1000 dilution) for 45 min at RT. Images were acquired with the Axiovert 200 M microscope (Zeiss, Oberkochen, Germany) and analyzed by ImageJ 1.54j (Bethesda, MD, USA). 

### 4.5. RNA Extraction and Quantitative Real-Time PCR

Total RNA from mouse cardiac tissue was isolated using RNA-Bee (Amsbio, Frankfurt am Main, Germany), and a quantitative real-time PCR was performed with Brilliant SYBR Green qPCR master mix (Applied Biosystems, Thermo Fischer Scientific, Darmstadt, Germany), as described [39]. The relative amount of target mRNA was determined using the comparative threshold (Ct) method, as previously described [39]. The mRNA contents of target genes were normalized to the expression of hypoxanthine phosphoribosyl transferase (HPRT) and ribosomal protein lateral stalk subunit P0 (RPLP0).

### 4.6. Protein Extraction and Immunoblotting

Left ventricle (LV) myocardium from male and female CVB3 mice was homogenized in Laemmli buffer (253 mM Tris/HCL pH 6.8, 8% SDS, 40% glycerin, 200 mM DTT, 0.4% bromophenol blue), as described [39]. Proteins were quantified using the BCA Assay (Pierce Protein Biology, Thermo Fisher Scientific, Darmstadt, Germany). Equal amounts of total proteins were separated on SDS-polyacrylamide gels and transferred to a nitrocellulose membrane. The membranes were immunoblotted overnight with the following primary antibodies: ERK (1:1000) p-ERK (1:2000), p38 (1:500), p-p38 (1:500), AKT (1:500), pAKT (1:500), Col1A1 (1:400), Col3A1 (1:400), NFκBp65 (1:200), TLR4 (1:500) (all: Santa Cruz Biotechnology, Heidelberg, Germany) and IL-6 (1:1000, Cell Signaling, Leiden, The Netherlands). Equal sample loading was confirmed by analysis of actin (1:1500, Santa Cruz Biotechnology, Heidelberg, Germany, USA) or HSP60 (1:1000, Cell Signaling, Leiden, The Netherlands). Immunoreactive proteins were detected using ECL Plus (GE Healthcare, Munich, Germany) and quantified with ImageLab (Bio-Rad Laboratories, Feldkirchen, Germany).

### 4.7. Statistical Analysis

All data are given as means ± standard error of *mean* (SEM). The data were evaluated using the non-parametric Mann–Whitney test for two independent groups or by two-way ANOVA. Statistical analyses were performed with GraphPad Prism 10 (GraphPad Software, La Jolla, CA, USA). Statistical significance was accepted when *p* < 0.05.

## 5. Conclusions

Taken together, the present study’s results show a sex-specific expression of pro-inflammatory markers, as well as inflammation- and remodeling-related miRNAs, under basal conditions and after CVB3-induced myocarditis in murine hearts. In particular, the male sex-specific downregulation of anti-inflammatory miRNA after CVB3-induced myocarditis may be partly responsible for the reported sex differences in the inflammatory response leading to worse outcomes in males after viral myocarditis.

## Figures and Tables

**Figure 1 ijms-25-09666-f001:**
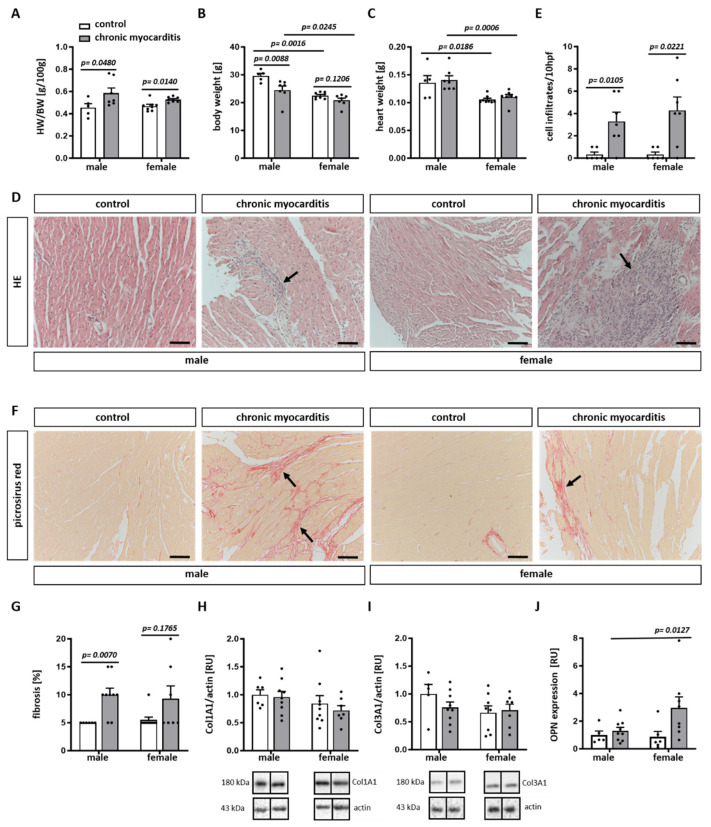
CVB3-infected mice show cardiac hypertrophy, increased numbers of immune cell infiltrates, and fibrosis in myocardial tissue. Heart to body weight ratio (HW/BW) (**A**), body weight (**B**), and heart weight (**C**) were measured 28 days after infection with CVB3. Representative images of HE-staining of myocardial tissue from male and female CVB3-infected mice (**D**). Magnification 200×. Arrows show cell infiltrates. Black scale bar = 100 µm. hpf = high power fields. Statistical analyses of myocardial cell infiltrates (**E**) in male and female CVB3-infected mice. Data are shown as the mean ± SEM (n = 5–10/group, represented as dots). Representative images of picrosirius red–dyed staining of cardiac tissue from male and female CVB3-infected mice (**F**). Magnification 200×. Arrows show fibrotic area. Black scale bar = 100 µm. Corresponding statistics of fibrosis in CVB3-infected mice (**G**). Data are shown as the mean ± SEM (n = 5–10/group). Western blot expression analyses of Col1A1 (**H**) and Col3A1 (**I**) performed with cardiac tissue lysates from CVB3-infected (chronic myocarditis) and non-infected (control) male and female mice. Data are shown as means ± SEM (n = 5–9). Real-time PCR analysis of OPN (**J**) mRNA expression in hearts from CVB3-infected (chronic myocarditis) versus non-infected (control) male and female mice. Data are shown as means ± SEM. Western blot data were expressed in relative units [RU] (n = 5–9/group, represented as dots). Western blot data were normalized to the male control. Statistical test: Mann–Whitney test and two-way ANOVA with Bonferroni correction.

**Figure 2 ijms-25-09666-f002:**
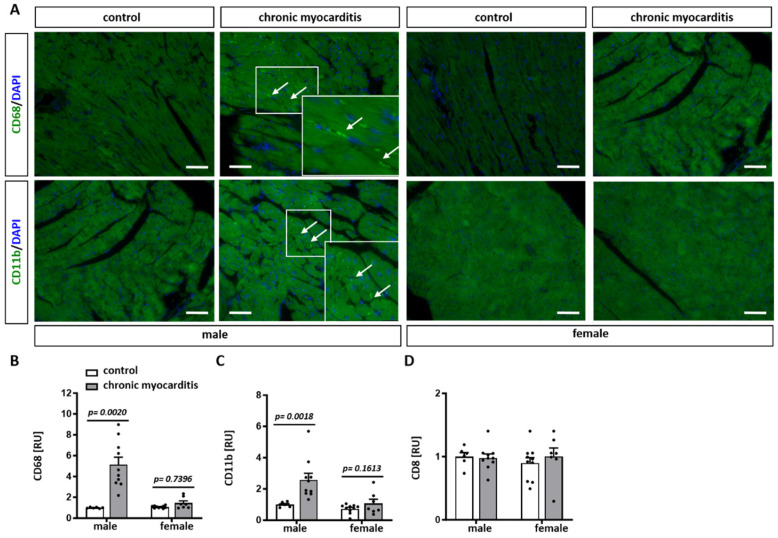
Accumulation of immune cell infiltrates in myocardial tissue of CVB3-infected mice. Representative images of CD68 and CD11b (green) expression in myocardial tissue from male and female control or CVB3-infected mice (**A**). DAPI (blue) counteracts for cell nuclei. Magnification 20×. Arrows show cell infiltrates. Scale bar = 100 µm. Statistical analyses of CD68 positive cells (**B**), CD11b (**C**) and CD8 (**D**) in the heart tissue of male and female control or CVB3-infected mice. Data are shown as the means ± SEM (n = 5–10/group, represented as dots).

**Figure 3 ijms-25-09666-f003:**
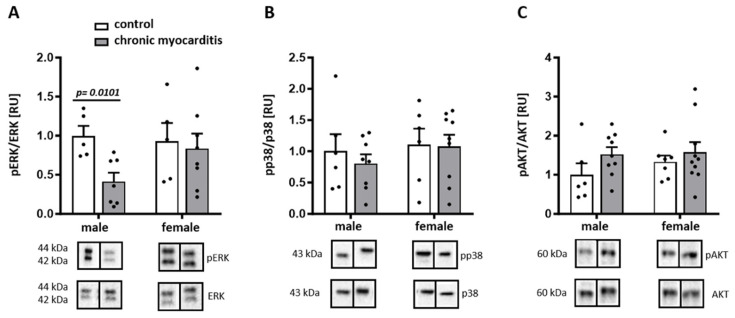
Sex differences in the expression and phosphorylation of MAP-kinases and AKT in a CVB3 chronic myocarditis mouse model. Western blot expression analysis of phosphorylated and total ERK (**A**), p38 (**B**), and AKT (**C**) performed with cardiac tissue lysates from CVB3-infected (chronic myocarditis) and non-infected (control) male and female mice. The phosphorylated proteins were normalized to the total proteins. Data are shown as means ± SEM (n = 5–9/group, represented as dots). All data were normalized to the male control and expressed in relative units [RU]. Statistical test: Mann–Whitney test.

**Figure 4 ijms-25-09666-f004:**
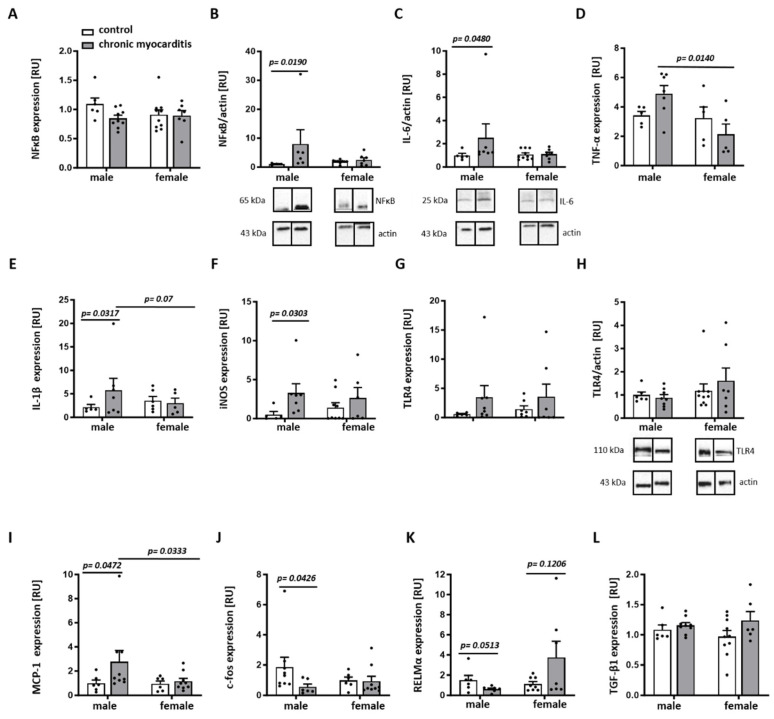
Sex differences in the expression of pro-inflammatory mediators in a CVB3 myocarditis mouse model. Real-time PCR analyses of NFκB (**A**), TNF-α (**D**), IL-1β (**E**), iNOS (**F**), TLR4 (**G**), MCP-1 (**I**), c-fos (**J**), RELMα (**K**), and TGF-β1 (**L**) mRNA expression in hearts from CVB3-infected (chronic myocarditis) versus non-infected (control) male and female mice. Western blot expression analysis of NFκB (**B**), IL-6 (**C**), and TLR4 (**H**) performed with cardiac tissue lysates from CVB3-infected (chronic myocarditis) and non-infected (control) male and female mice. Data are shown as means ± SEM (n = 5–10/group, represented as dots). Western blot data were normalized to the male control and expressed in relative units [RU]. Statistical test: Mann–Whitney test and two-way ANOVA with Bonferroni correction.

**Figure 5 ijms-25-09666-f005:**
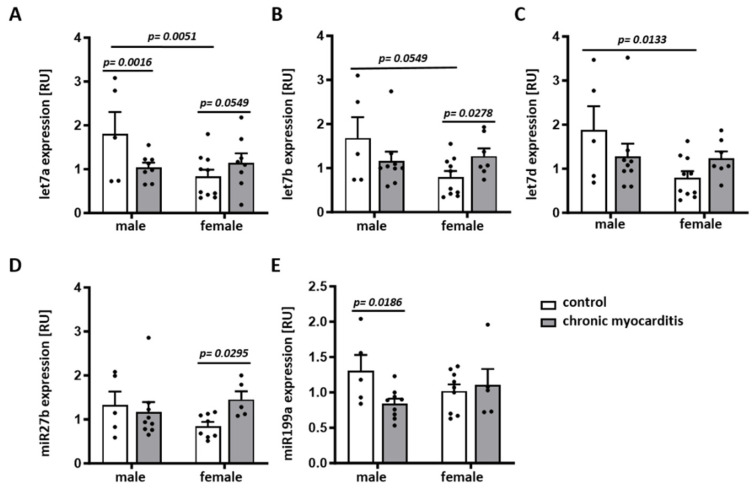
Sex differences in the expression of miRNAs in a CVB3 myocarditis mouse model. Real-time PCR analyses of let7a (**A**), let7b (**B**), let7d (**C**), miR27b (**D**), and miR199a (**E**) miRNA in hearts from CVB3-infected (chronic myocarditis) versus non-infected (control) male and female mice. Data are shown as means ± SEM (n = 5–10/group, represented as dots). Statistical test: Mann–Whitney test and two-way ANOVA with Bonferroni correction.

## Data Availability

The original contributions presented in the study are included in the article/Appendix A. Further inquiries can be directed to the corresponding author.

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
