# Peer review of "Sex Differences in Expression of Pro-Inflammatory Markers and miRNAs in a Mouse Model of CVB3 Myocarditis"

_ijms, 2024, doi:10.3390/ijms25179666_

Round 1
Reviewer 1 Report (Previous Reviewer 1)
Comments and Suggestions for Authors
The manuscript is well revised.
Author Response
Thank you for your suggestions for improvement.
Reviewer 2 Report (New Reviewer)
Comments and Suggestions for Authors
In this study, it was found that heart-passaged CVB3 induces inflammatory heart disease similar to the disease observed in clinical practice. Therefore, the authors investigated sex differences in the expression of pro-inflammatory markers and miRNAs.
Major:
Although this paper examines sex differences, it does not mention any variations in hormonal secretion or effects other than the cardioprotective effect of estrogen. Additionally, sex chromosomes were not discussed at all. The methods need to be presented in a more logical organizational sequence, with less specific experiments described before more sophisticated ones.
Minor:
Change "Cardiac mouse tissue" to "mouse cardiac tissue" as in line 24, remove the space in line 105. define the acronym "SEM" as used in line 162.
Author Response
Major:
Although this paper examines sex differences, it does not mention any variations in hormonal secretion or effects other than the cardioprotective effect of estrogen. Additionally, sex chromosomes were not discussed at all.
Response: This issue has been partly addressed in the introduction and now additionally in the discussion: “A variety of factors may contribute to the sex difference in the pathogenesis of CVB3-induced myocarditis (for review see PMID: 36937911). Gonadal hormones seem to play here primary role. In particular, gonadectomy in mice suppressed male, but increased female, susceptibility to CVB3 (PMID: 21806829). The protective effects of the female hormone estradiol and estrogen receptors due to the affecting pro- and anti-inflammatory pathways in immune cells have been previously shown (13, 14). In contrast to estrogen, the male sex hormone testosterone may contribute to the worse myocarditis outcome in males (PMID: 30638108). Finally, sex chromosomes may also affect the CVB3 pathogenesis (PMID: 21806829, PMID: 22384388). Although the role of cell sex was poorly understood, the involvement of Midline 1, TLR7 and TLR8 genes localized on the X-chromosome in immune response has been suggested (PMID: 35909127, PMID: 30276444, PMID: 24500834).”
The methods need to be presented in a more logical organizational sequence, with less specific experiments described before more sophisticated ones.
Response: We have changed the sequence of the methods.
Minor:
Change "Cardiac mouse tissue" to "mouse cardiac tissue" as in line 24, remove the space in line 105. define the acronym "SEM" as used in line 162.
Response: Thank you for your advice. We have made the corrections.
Round 2
Reviewer 2 Report (New Reviewer)
Comments and Suggestions for Authors
I have no further objections.
This manuscript is a resubmission of an earlier submission. The following is a list of the peer review reports and author responses from that submission.
Round 1
Reviewer 1 Report
Comments and Suggestions for Authors
The manuscript is very well presented and the conclusions are useful for understanding the pathophysiology of viral-induced myocarditis.
The sex-specific differences in the inflammatory response and regulation of the inflammation- and remodeling-related miRNAs in a mouse model of chronic viral myocarditis. The paper addresses sex differences in the miRNA expression in chronic myocarditis.
The male sex-specific downregulation of anti-inflammatory miRNA after CVB3-induced myocarditis may be partly responsible for the reported sex differences in the inflammatory response leading to worse outcomes in males after viral myocarditis.
The methodology is well-addressed. No specific improvements should be considered.
The conclusions are consistent with the evidence and arguments presented. All main questions posed were addressed by specific experiments: Male and female ABY/SnJ mice were infected with CVB3. 28 days later, cardiac tissue from both infected and control mice was used for real-time PCR and western blot analysis. NFκB, IL-6, iNOS, TNF-α, IL-1β, MCP-1, c-fos, and osteopontin (OPN) were used to examine the inflammatory state in the heart. Furthermore, the expression of several inflammation- and remodeling-related miRNAs was analyzed
I have no additional comments on the tables, figures, and quality of the data.
Author Response
Reviewers' Comments to Author:
Reviewer 2: In this article, the authors investigating sex differences in a mouse model of CVB3-induced myocarditis. Despite the high relevance of sex differences in heart failure worldwide and some interesting findings, this article cannot be accepted for publication in its current version.
Major concerns:
The study is only descriptive. Functional analysis of the mice are missing. The argument of housing conditions is not appropriate in this context. The authors need to demonstrate that they have a model of DCM, at least with ex vivo functional analysis. Especially, with respect to the last paragraph. The authors write, “miRNAs are involved in the regulation of cardiac function and the immune response/inflammation (line 230)”. The authors have to prove this.
- Thank you for the comment. Unfortunately, since the mice must be housed in the S2 animal facility, we were not able to perform echocardiography analyses. We included this point in the limitations of the study.
Furthermore, not only the functional characterization is missing, but also the characterization of the immune response using FACS. It is not sufficient to evaluate only the total immune infiltrates in the heart. Further characterization of these immune cells can be done, for example, by immunohistochemistry.
- We thank the reviewer for raising this important issue. We agree that we should characterize the immune response using FACS. However, in Germany the authorities appeal to minimize animal studies. The local authorities at the Charité encouraged us to follow the 3R guidelines (replacement, reduction, and refinement). We addressed this issue in the limitations of the study.
In addition, there are already publications on sex differences in CVB3-induced myocarditis, e.g:
https://pubmed.ncbi.nlm.nih.gov/36741845/
https://pubmed.ncbi.nlm.nih.gov/37662585/
https://pubmed.ncbi.nlm.nih.gov/38196574/
https://pubmed.ncbi.nlm.nih.gov/34445539/
https://pubmed.ncbi.nlm.nih.gov/38146431/
https://pubmed.ncbi.nlm.nih.gov/37365150/
This means that the entire study lacks novelty. Although there were some interesting aspects, additional in vitro and/or blocking experiments (e.g. of the relevant miRNAs) are missing to gain more substantial insights.
- We agree that several previous reports, including those referred by the reviewer, suggest sex differences in the pathogenesis of myocarditis comprising mitochondrial biology and inflammation. Now, the current study analyzing the inflammation- and remodeling-related miRNAs provides a potential explanation for the previously observed differences. Therefore, we believe that the study provides novel information.
Additional comments to the authors:
- There are several typos in the manuscript e.g. non-immunized mice instead of non-infected mice. The authors infected the mice; they did not immunize the mice.
- We thank for the observation. We reviewed the manuscript and made corrections if necessary.
- Method section:
The material and methods are not appropriate reported. Animal studies should be reported according the ARRIVE guidelines, including information regarding randomization, final n-numbers for each investigation per group (not only the total n-number of animals which are infected or not), housing conditions (including facility names), anesthesia for cervical dislocation etc.
- Thank you for the comment. We addressed this point and added the missing information.
Where was the virus purchased from?
- The virus was a gift from the Institute of Biochemistry of the Charité-Universitätsmedizin
The authors stated “All procedures were performed in accordance with the Guide for the Care and Use of Laboratory Animals published by the US National Institutes of Health and were approved by local authorities (Landesamt für Gesundheit und Soziales, Berlin).“ Since Germany is part of the European Union, it must be assumed that the experiments are carried out in accordance with the European legislation for the Care and Use of Laboratory Animals (Directive 2010/63/EU). This statement is therefore incorrect and must be corrected. Furthermore, the number of the approval needs to be added.
- Thank you for the observation. We corrected this issue and added the information.
It is not clear which housekeeping gene was ultimately used to normalize the mRNA data. Two housekeeping genes are mentioned in the methods section (HPRT and RPLP0). Especially when comparing male and female animals, the same housekeeping gene should be used.
- We used two house keeping genes to corroborate our results. For the results presented in this manuscript we used the same housekeeping gene. The results were normalized to HPRT.
Furthermore, normalization to male controls does not seem logical. The absolute expression would be more appropriate, also to better interpret the differences between male and female mice. In addition, forward and reverse primers should be listed. Simply adding a reference does not correspond to the state of the art.
- We now show the absolute expression of the mRNA andlisted the primers in a new table 1.
How were hypertrophy, immune cell infiltration and fibrosis scored? Did the authors establish a scoring system or how exactly was this done? This is not clear from the paragraph.
- The overall fibrosis, muscle hypertrophy and immune cell infiltrates were determined via semiquantitative, visual evaluation. All sections were blindly evaluated by three different investigators. We already published this method in PMID: 34122450 and PMID: 37365150
We addressed this issue in material and methods.
What do the authors mean by hpf?
- Positive cells were quantified in 10 high power fields (hpf) (field of vision in x200 original magnification. We include this information in the manuscript.
- Results/figures:
The authors should be more precise in their wording. Some headings or sentences are exaggerated, e.g. line 183 "Sex differences in the activation of ERK in the mouse model of CVB3-induced chronic myocarditis". Here, no differences were found between males and females. Only pERK/ERK was downregulated in male CVB3-infected mice compared to male controls. The title "Sex-dependent activation of ERK in the mouse model of CVB3-induced chronic myocarditis" would be more precise.
- Thank you for the comment. We made the correction.
In general, the figure legends should be improved. Important information like final n-numbers, statistical analysis, and abbreviations should be added.
- We thank for the comment. The figure legends were reviewed and modified where necessary.
In figure 1 (and in the supplement) blots for Col1a1 and Col3a1 are missing. Also information about the scoring (immune cell infiltrates and fibrosis) need to be added.
- Thank you for the comment. The blots were added. The amount of immune cell infiltrates and fibrosis was evaluated in 10 high power fields (magnification x200) and then averaged.
In general, one would not expect hypertrophy as a sign of DCM. We also know from our own studies that body weight and heart weight decreased in CVB3-infected mice. Logically, therefore, the mice show no signs of hypertrophy. This decrease in body and heart weight can also be observed at day 10 after infection (https://pubmed.ncbi.nlm.nih.gov/38146431/) In the aforementioned publication, the authors already showed sex differences between male and female infected mice and also a reduction in wall thicknesses. The authors also mentioned that a hypertrophy score was determined. This data cannot be found in Figure 1 or is missing there.
- Thanks for the observation. We added the missing information in figure 1.
Line 212: Why two p-values were reported here?
- We thank for the annotation. It was corrected.
Sometimes there is an inconsistency between the referring to the figures (e.g., line 164 versus line 193).
- We thank for the comment. It was corrected.

Reviewer 2 Report
Comments and Suggestions for Authors
Title:
Sex differences in expression of pro-inflammatory markers and miRNAs in the mouse model of CVB3 myocarditis
ID: ijms-2863095
Authors: Misael Estepa, Maximilian H. Niehues, Natalie Haritonow, Ursula Müller-Werdan, Yury Ladilov, Maria Luisa Barcena, Vera Regitz-Zagrosek
In this article, the authors investigating sex differences in a mouse model of CVB3-induced myocarditis. Despite the high relevance of sex differences in heart failure worldwide and some interesting findings, this article cannot be accepted for publication in its current version.
Major concerns:
The study is only descriptive. Functional analysis of the mice are missing. The argument of housing conditions is not appropriate in this context. The authors need to demonstrate that they have a model of DCM, at least with ex vivo functional analysis. Especially, with respect to the last paragraph. The authors write, “miRNAs are involved in the regulation of cardiac function and the immune response/inflammation (line 230)”. The authors have to prove this. Furthermore, not only the functional characterization is missing, but also the characterization of the immune response using FACS. It is not sufficient to evaluate only the total immune infiltrates in the heart. Further characterization of these immune cells can be done, for example, by immunohistochemistry.
In addition, there are already publications on sex differences in CVB3-induced myocarditis, e.g:
· https://pubmed.ncbi.nlm.nih.gov/36741845/
· https://pubmed.ncbi.nlm.nih.gov/37662585/
· https://pubmed.ncbi.nlm.nih.gov/38196574/
· https://pubmed.ncbi.nlm.nih.gov/34445539/
· https://pubmed.ncbi.nlm.nih.gov/38146431/
· https://pubmed.ncbi.nlm.nih.gov/37365150/
This means that the entire study lacks novelty. Although there were some interesting aspects, additional in vitro and/or blocking experiments (e.g. of the relevant miRNAs) are missing to gain more substantial insights.
Additional comments to the authors:
1. There are several typos in the manuscript e.g. non-immunized mice instead of non-infected mice. The authors infected the mice; they did not immunize the mice.
2. Method section:
- The material and methods are not appropriate reported. Animal studies should be reported according the ARRIVE guidelines, including information regarding randomization, final n-numbers for each investigation per group (not only the total n-number of animals which are infected or not), housing conditions (including facility names), anesthesia for cervical dislocation etc.
- Where was the virus purchased from?
- The authors stated “All procedures were performed in accordance with the Guide for the Care and Use of Laboratory Animals published by the US National Institutes of Health and were approved by local authorities (Landesamt für Gesundheit und Soziales, Berlin).“ Since Germany is part of the European Union, it must be assumed that the experiments are carried out in accordance with the European legislation for the Care and Use of Laboratory Animals (Directive 2010/63/EU). This statement is therefore incorrect and must be corrected. Furthermore, the number of the approval needs to be added.
- It is not clear which housekeeping gene was ultimately used to normalize the mRNA data. Two housekeeping genes are mentioned in the methods section (HPRT and RPLP0). Especially when comparing male and female animals, the same housekeeping gene should be used. Furthermore, normalization to male controls does not seem logical. The absolute expression would be more appropriate, also to better interpret the differences between male and female mice. In addition, forward and reverse primers should be listed. Simply adding a reference does not correspond to the state of the art.
- How were hypertrophy, immune cell infiltration and fibrosis scored? Did the authors establish a scoring system or how exactly was this done? This is not clear from the paragraph. What do the authors mean by hpf?
3. Results/figures:
- The authors should be more precise in their wording. Some headings or sentences are exaggerated, e.g. line 183 "Sex differences in the activation of ERK in the mouse model of CVB3-induced chronic myocarditis". Here, no differences were found between males and females. Only pERK/ERK was downregulated in male CVB3-infected mice compared to male controls. The title "Sex-dependent activation of ERK in the mouse model of CVB3-induced chronic myocarditis" would be more precise.
- In general, the figure legends should be improved. Important information like final n-numbers, statistical analysis, and abbreviations should be added.
- In figure 1 (and in the supplement) blots for Col1a1 and Col3a1 are missing. Also information about the scoring (immune cell infiltrates and fibrosis) need to be added.
- In general, one would not expect hypertrophy as a sign of DCM. We also know from our own studies that body weight and heart weight decreased in CVB3-infected mice. Logically, therefore, the mice show no signs of hypertrophy. This decrease in body and heart weight can also be observed at day 10 after infection (https://pubmed.ncbi.nlm.nih.gov/38146431/) In the aforementioned publication, the authors already showed sex differences between male and female infected mice and also a reduction in wall thicknesses. The authors also mentioned that a hypertrophy score was determined. This data cannot be found in Figure 1 or is missing there.
- Line 212: Why two p-values were reported here?
- Sometimes there is an inconsistency between the referring to the figures (e.g., line 164 versus line 193).

Comments on the Quality of English Languagesee above
Author Response

(The authors gave the same response as above.)

Round 2
Reviewer 2 Report
Comments and Suggestions for Authors
Although some of my comments have been addressed, there are still important comments that have not been adequately addressed. Therefore, the manuscript cannot be accepted for publication in its current form.
Major concerns:
The study is only descriptive. Functional analysis of the mice are missing. The argument of housing conditions is not appropriate in this context. The authors need to demonstrate that they have a model of DCM, at least with ex vivo functional analysis. Especially, with respect to the last paragraph. The authors write, “miRNAs are involved in the regulation of cardiac function and the immune response/inflammation (line 230)”. The authors have to prove this.
Answer from the authors:
- Thank you for the comment. Unfortunately, since the mice must be housed in the S2 animal facility, we were not able to perform echocardiography analyses. We included this point in the limitations of the study.
Respond from the reviewer:
- As I have already mentioned, cardiac function can also be investigated by ex vivo analyses, e.g. on frozen tissue. Functional confirmation of the DCM model is still missing and necessary to evaluate the miRNA findings.
Furthermore, not only the functional characterization is missing, but also the characterization of the immune response using FACS. It is not sufficient to evaluate only the total immune infiltrates in the heart. Further characterization of these immune cells can be done, for example, by immunohistochemistry.
Answer from the authors:
- We thank the reviewer for raising this important issue. We agree that we should characterize the immune response using FACS. However, in Germany the authorities appeal to minimize animal studies. The local authorities at the Charité encouraged us to follow the 3R guidelines (replacement, reduction, and refinement). We addressed this issue in the limitations of the study.
Respond from the reviewer:
- As mentioned above, cardiac immune cell infiltration can also be detected by immunohistochemistry. As the authors have shown representative images of tissue sections, these samples, which allow staining of immune cells on tissue sections, are already available. So no further study would be necessary.
In addition, there are already publications on sex differences in CVB3-induced myocarditis. This means that the entire study lacks novelty. Although there were some interesting aspects, additional in vitro and/or blocking experiments (e.g. of the relevant miRNAs) are missing to gain more substantial insights.
Answer from the authors:
- We agree that several previous reports, including those referred by the reviewer, suggest sex differences in the pathogenesis of myocarditis comprising mitochondrial biology and inflammation. Now, the current study analyzing the inflammation- and remodeling-related miRNAs provides a potential explanation for the previously observed differences. Therefore, we believe that the study provides novel information.
Respond from the reviewer:
- As I mentioned in the first overview report, blocking experiments (e.g. of the relevant miRNAs) are necessary to prove the hypothesis and gain new insights. Otherwise, the novelty is still too weak.
Additional comments to the authors:
- Where was the virus purchased from?
Answer from the authors:
- The virus was a gift from the Institute of Biochemistry of the Charité-Universitätsmedizin Berlin.
Respond from the reviewer:
- Please be more specific here. If the people who kindly provided the virus are not named as authors, they should at least be named in the relevant section.
- It is not clear which housekeeping gene was ultimately used to normalize the mRNA data. Two housekeeping genes are mentioned in the methods section (HPRT and RPLP0). Especially when comparing male and female animals, the same housekeeping gene should be used. Furthermore, normalization to male controls does not seem logical. The absolute expression would be more appropriate, also to better interpret the differences between male and female mice. In addition, forward and reverse primers should be listed. Simply adding a reference does not correspond to the state of the art.
Answer from the authors:
- We used two house keeping genes to corroborate our results. For the results presented in this manuscript we used the same housekeeping gene. The results were normalized to HPRT.
- We now show the absolute expression of the mRNA and listed the primers in a new table 1.
Respond from the reviewer:
- Some errors have occurred in the updated illustrations. They are not displayed correctly in the manuscript.
- How were hypertrophy, immune cell infiltration and fibrosis scored? Did the authors establish a scoring system or how exactly was this done? This is not clear from the paragraph. What do the authors mean by hpf?
Answer from the authors:
- The overall fibrosis, muscle hypertrophy and immune cell infiltrates were determined via semiquantitative, visual evaluation. All sections were blindly evaluated by three different investigators. We already published this method in PMID: 34122450 and PMID: 37365150. We addressed this issue in material and methods.
Respond from the reviewer:
- Some errors have occurred in the updated illustrations. They are not displayed correctly in the manuscript. The hypertrophy score or the collagen blots are not yet included in the version uploaded to the system. Therefore, no evaluation of the correction is possible.
- In general, one would not expect hypertrophy as a sign of DCM. We also know from our own studies that body weight and heart weight decreased in CVB3-infected mice. Logically, therefore, the mice show no signs of hypertrophy. This decrease in body and heart weight can also be observed at day 10 after infection (https://pubmed.ncbi.nlm.nih.gov/38146431/) In the aforementioned publication, the authors already showed sex differences between male and female infected mice and also a reduction in wall thicknesses. The authors also mentioned that a hypertrophy score was determined. This data cannot be found in Figure 1 or is missing there.
Answer from the authors:
- Thanks for the observation. We added the missing information in figure 1
Respond from the reviewer:
- Some errors have occurred in the updated illustrations. They are not displayed correctly in the manuscript. The hypertrophy score or the collagen blots are not yet included in the version uploaded to the system. Therefore, no evaluation of the correction is possible.
- Furthermore, the statement of hypertrophy and DCM needs to be further/better discussed.
Comments on the Quality of English Languagesee above
Author Response
Major concerns:
- The study is only descriptive. Functional analysis of the mice are missing. The argument of housing conditions is not appropriate in this context. The authors need to demonstrate that they have a model of DCM, at least with ex vivo functional analysis. Especially, with respect to the last paragraph. The authors write, “miRNAs are involved in the regulation of cardiac function and the immune response/inflammation (line 230)”. The authors have to prove this.
Answer from the authors:
Thank you for the comment. Unfortunately, since the mice must be housed in the S2 animal facility, we were not able to perform echocardiography analyses. We included this point in the limitations of the study.
Reviewer’s response:
As I have already mentioned, cardiac function can also be investigated by ex vivo analyses, e.g. on frozen tissue. Functional confirmation of the DCM model is still missing and necessary to evaluate the miRNA findings.
Answer from the authors:
We apologize for the previous misunderstanding of the reviewer's request. Probably under “functional confirmation of the DCM model on frozen tissue” the reviewer means the morphological characterization of the model. Unfortunately, the quality of the tissue sections does not allow an accurate assessment of the morphology of the left ventricle.
Instead of morphology, the myocarditis model was characterized by the immune infiltrates. In the revised manuscript, this issue has been further addressed and the data are presented in the new Figure 2.
- Furthermore, not only the functional characterization is missing, but also the characterization of the immune response using FACS. It is not sufficient to evaluate only the total immune infiltrates in the heart. Further characterization of these immune cells can be done, for example, by immunohistochemistry.
Answer from the authors:
We thank the reviewer for raising this important issue. We agree that we should characterize the immune response using FACS. However, in Germany the authorities appeal to minimize animal studies. The local authorities at the Charité encouraged us to follow the 3R guidelines (replacement, reduction, and refinement). We addressed this issue in the limitations of the study.
Reviewer’s response:
As mentioned above, cardiac immune cell infiltration can also be detected by immunohistochemistry. As the authors have shown representative images of tissue sections, these samples, which allow staining of immune cells on tissue sections, are already available. So no further study would be necessary.
Answer from the authors:
As recommended by the reviewer we further characterized the immune cell infiltrates. These data are presented now in the new Figure 2.
- In addition, there are already publications on sex differences in CVB3-induced myocarditis. This means that the entire study lacks novelty. Although there were some interesting aspects, additional in vitro and/or blocking experiments (e.g. of the relevant miRNAs) are missing to gain more substantial insights.
Answer from the authors:
We agree that several previous reports, including those referred by the reviewer, suggest sex differences in the pathogenesis of myocarditis comprising mitochondrial biology and inflammation. Now, the current study analyzing the inflammation- and remodeling-related miRNAs provides a potential explanation for the previously observed differences. Therefore, we believe that the study provides novel information.
Reviewer’s response:
As I mentioned in the first overview report, blocking experiments (e.g. of the relevant miRNAs) are necessary to prove the hypothesis and gain new insights. Otherwise, the novelty is still too weak.
Answer from the authors:
We thank the reviewer for the comment. However, we believe that suggested “blocking experiments” requiring a new mice group will be a topic of the follow-up study. For the current project unlikely the local authorities at the Charité encouraging us to follow the 3R guidelines will allow any additional experiments.
Additional comments to the authors:
- Where was the virus purchased from?
Answer from the authors:
The virus was a gift from the Institute of Biochemistry of the Charité-Universitätsmedizin Berlin.
Reviewer’s response:
Please be more specific here. If the people who kindly provided the virus are not named as authors, they should at least be named in the relevant section.
Answer from the authors:
We included the names of the people who kindly provided the virus in the acknowledgments.
- It is not clear which housekeeping gene was ultimately used to normalize the mRNA data. Two housekeeping genes are mentioned in the methods section (HPRT and RPLP0). Especially when comparing male and female animals, the same housekeeping gene should be used. Furthermore, normalization to male controls does not seem logical. The absolute expression would be more appropriate, also to better interpret the differences between male and female mice. In addition, forward and reverse primers should be listed. Simply adding a reference does not correspond to the state of the art.
Answer from the authors:
We used two housekeeping genes to corroborate our results. For the results presented in this manuscript, we used the same housekeeping gene. The results were normalized to HPRT.
We now show the absolute expression of the mRNA and list the primers in a new table 1.
- How were hypertrophy, immune cell infiltration and fibrosis scored? Did the authors establish a scoring system or how exactly was this done? This is not clear from the paragraph. What do the authors mean by hpf?
Answer from the authors:
The overall fibrosis, muscle hypertrophy and immune cell infiltrates were determined via semiquantitative, visual evaluation. All sections were blindly evaluated by three different investigators. We already published this method in PMID: 34122450 and PMID: 37365150. We addressed this issue in material and methods.
- In general, one would not expect hypertrophy as a sign of DCM. We also know from our own studies that body weight and heart weight decreased in CVB3-infected mice. Logically, therefore, the mice show no signs of hypertrophy. This decrease in body and heart weight can also be observed at day 10 after infection (https://pubmed.ncbi.nlm.nih.gov/38146431/) In the aforementioned publication, the authors already showed sex differences between male and female infected mice and also a reduction in wall thicknesses. The authors also mentioned that a hypertrophy score was determined. This data cannot be found in Figure 1 or is missing there.
Answer from the authors:
Thanks for the observation. We added the missing information in figure 1
Reviewer’s response:
Some errors have occurred in the updated illustrations. They are not displayed correctly in the manuscript. The hypertrophy score or the collagen blots are not yet included in the version uploaded to the system. Therefore, no evaluation of the correction is possible.
Furthermore, the statement of hypertrophy and DCM needs to be further/better discussed.
Answer from the authors:
We apologize for this. We have now corrected the updated Figures.
Furthermore, the hypertrophy and DCM were addressed in the section " Limitations of the study".